# Administration of Warfarin Inhibits the Development of Cerulein-Induced Edematous Acute Pancreatitis in Rats

**DOI:** 10.3390/biom13060948

**Published:** 2023-06-06

**Authors:** Katarzyna Konarska-Bajda, Piotr Ceranowicz, Jakub Cieszkowski, Grzegorz Ginter, Agnieszka Stempniewicz, Krystyna Gałązka, Beata Kuśnierz-Cabala, Paulina Dumnicka, Joanna Bonior, Zygmunt Warzecha

**Affiliations:** 1Department of Physiology, Faculty of Medicine, Jagiellonian University Medical College, 31-531 Kraków, Poland; konarskake@gmail.com (K.K.-B.); jakub.cieszkowski@uj.edu.pl (J.C.); grzegorz.ginter@uj.edu.pl (G.G.); agnieszka.stempniewicz@doctoral.uj.edu.pl (A.S.); 2Department of Pediatric Cardiology, University Children’s Hospital in Cracow, 30-663 Kraków, Poland; 3Department of Pathology, Faculty of Medicine, Jagiellonian University Medical College, 31-531 Kraków, Poland; krystyna.galazka@uj.edu.pl; 4Chair of Clinical Biochemistry/Chair of Medical Biochemistry, Jagiellonian University Medical College, 31-034 Kraków, Poland; beata.kusnierz-cabala@uj.edu.pl (B.K.-C.);; 5Department of Medical Physiology, Faculty of Health Sciences, Jagiellonian University Medical College, 31-126 Kraków, Poland; joanna.bonior@uj.edu.pl

**Keywords:** acute pancreatitis, coumarin, warfarin, inflammation, coagulation

## Abstract

Acute pancreatitis (AP) is a severe disease with high morbidity and mortality in which inflammation and coagulation play crucial roles. The development of inflammation leads to vascular injury, endothelium and leukocytes stimulation, and an increased level of tissue factor, which results in the activation of the coagulation process. For this reason, anticoagulants may be considered as a therapeutic option in AP. Previous studies have shown that pretreatment with heparin, low-molecular-weight heparin (LMWH), or acenocoumarol inhibits the development of AP. The aim of the present study was to check if pretreatment with warfarin affects the development of edematous pancreatitis evoked by cerulein. Warfarin (90, 180, or 270 µg/kg/dose) or saline were administered intragastrically once a day for 7 days consecutively before the induction of AP. AP was evoked by the intraperitoneal administration of cerulein. The pre-administration of warfarin at doses of 90 or 180 µg/kg/dose reduced the histological signs of pancreatic damage in animals with the induction of AP. Additionally, other parameters of AP, such as an increase in the serum activity of lipase and amylase, the plasma concentration of D-dimer, and interleukin-1β, were decreased. In addition, pretreatment with warfarin administered at doses of 90 or 180 µg/kg/dose reversed the limitation of pancreatic blood flow evoked by AP development. Warfarin administered at a dose of 270 µg/kg/dose did not exhibit a preventive effect in cerulein-induced AP. Conclusion: Pretreatment with low doses of warfarin inhibits the development of AP evoked by the intraperitoneal administration of cerulein.

## 1. Introduction

Inflammation and coagulation play an essential role in the pathogenesis of many diseases [1,2,3,4,5], and there is an increasing number of evidence for extensive cross-talk between those two processes [6,7]. Inflammation leads to the activation of coagulation, which in turn enhances the inflammatory process on the basis of positive feedback. The same mechanism of mutually reinforcing inflammation and coagulation also occurs when coagulation is initially stimulated [8,9]. The role of pro-inflammatory cytokines in the activation of coagulation cascade involves, among others, promoting the expression of tissue factor on monocytes and endothelium, leading to the formation of thrombin and fibrin [10,11,12,13]. Besides the promotion of clot formation, inflammation also reduces the activity of anticoagulant mechanisms and inhibits fibrinolysis [10,14]. Infection and inflammation can activate coagulation and, in severe cases, may result in disseminated intravascular coagulation (DIC) [15]. These mechanisms are also implicated in the pathogenesis of organ dysfunction in patients with sepsis [16].

It should also be noted that the activation of coagulation does not only lead to clot formation [17,18] but it also simultaneously activates the mechanisms of the inflammatory process [8,9]. The tissue factor—the factor VII complex also can bind to protease-activated receptors (PARs) [19,20]. These receptors are found on platelets, endothelial cells, monocytes, lymphocytes, macrophages, mast cells, and dendritic cells [21]. The stimulation of PARs leads to the activation of these cells and the production of free radicals, increased expression of adhesion molecules by endothelial cells, activation of complement, and release of cytokines and chemokines [9,19,22]. In addition, the stimulation of PARs in platelets results in the release of soluble ligand for the CD-40 receptor (sCD40L) [23]. CD40 modulates T-cell-mediated effector functions and general immune response; it also promotes the expression of proinflammatory cytokines, adhesion molecules, and matrix-degrading activities [24,25]. Moreover, thrombin stimulates the production of interleukin 6 (IL-6) and monocyte chemotactic protein (MCP-1) in fibroblasts, mononuclear cells, and epithelial cells [14].

Coagulative disorders are also observed in AP; they are related to the severity of this disease [26,27,28,29,30,31]. In mild AP, coagulation disorders take the form of local scattered intravascular thrombosis in the pancreatic circulation [32]. In severe AP, the activation of coagulation can result in DIC and is associated with poor prognosis [33,34]. Several laboratory markers of inflammation and hemostasis were described as useful in the prediction of AP severity and its mortality [6,29,35,36].

Numerous previously performed experimental studies have shown that that administration of unfractionated heparin, a pleotropic glycosaminoglycan with anticoagulant properties and an anti-inflammatory effect [37], exhibits a protective and therapeutic effect in AP evoked by bile acid [38], cerulein [39], and pancreatic ischemia followed by reperfusion [40]. Additionally, previous clinical studies have shown that unfractionated heparin prevents the development of AP and/or has a therapeutic effect in AP caused by endoscopic retrograde cholangiopancreatography [41] and hypertriglyceridemia [42,43].

Similar protective and therapeutic effects in animal studies [44,45] and in clinical AP [46,47,48] have also been found after the administration of low-molecular-weight heparin (LMWH).

Activated protein C inhibits coagulation, inflammation, and cell death, as well as promotes fibrinolysis and reduces mortality in patients with severe sepsis [49]. Experimental animal studies have shown that the administration of activated protein C reduces pancreatic tissue damage, inhibits bacterial translocation to mesenteric lymph nodes, and decreases the serum level of inflammatory markers [45], as well as improves 24 h survival [50] in severe AP in rats. In addition, some studies indicate that the protective effect of activated protein C in severe AP involves the up-regulation of thrombomodulin expression [51]. However, clinical trials have not confirmed the effectiveness of activated protein C in the treatment of AP [52,53]. On the other hand, there is a clinical report which claims that treatment with recombinant human soluble thrombomodulin may prevent progression from pancreatic necrosis/ischemia to walled-off necrosis in patients with severe AP [48].

Coumarins are vitamin K antagonists. Coumarins inhibit the vitamin K epoxide reductase, an enzyme that reduces the oxidized form of vitamin K back to its active, reduced form. The reduced form of vitamin K is a necessary cofactor for hepatic γ-glutamyl carboxylase activity. This enzyme adds a carboxyl group to glutamic acid residues in immature precursors of clotting factors such as factor II, factor VII, factor IX, and factor X, as well as anticoagulant proteins S, C, and Z. Coumarins reduce the plasma level of vitamin-K-dependent mature clotting factors, leading to a reduction in blood coagulability [54]. Previous experimental studies have shown that acenocoumarol, a coumarin-derived and vitamin K antagonist, can prevent the development of inflammation and support the healing of the pancreas in the course of experimental AP [55,56,57]. Another coumarin derive, warfarin, has a more favorable pharmaceutical profile as an anticoagulant [58]. Our recent studies have shown that warfarin exhibits a protective and healing effect in AP evoked by pancreatic ischemia, followed by reperfusion [59,60]. The effect of pretreatment with warfarin on the development of AP evoked by a primary non-vascular mechanism has not been tested. Therefore, the aim of the present study was to investigate whether warfarin influences the development of cerulein-induced AP.

## 2. Materials and Methods

### 2.1. Animals and Treatment

All experiments were performed according to the protocol approved by the First Local Commission of Ethics for the Care and Use of Laboratory Animals in Cracow (Permits Number 25/2016 released on 20 July 2016, 95/2017 released on 20 December 2017, and 536/2021 released on 16 December 2021).

The research was conducted on 64 male Wistar rats weighing 200–220 g, which were kept in cages in a windowless colony room. The temperature in the colony room was adjusted at 22 ± 1 °C with relative humidity of 50 ± 10% and a 12/12 h light–dark photoperiod. During the study, the animals had free access to food and water. After a one-week period of acclimation to their new environment, the animals were divided randomly into eight equal groups:(1)Saline-treated control rats without induction of AP (Control—C);(2)Rats without induction of AP pretreated with warfarin administered at a dose of 90 µg/kg/dose (WF 90 + Saline);(3)Rats without induction of AP pretreated with warfarin administered at a dose of 180 µg/kg/dose (WF 180 + Saline);(4)Rats without induction of AP pretreated with warfarin administered at a dose of 270 µg/kg/dose (WF 270 + Saline);(5)Rats with cerulein-induced AP pretreated with saline (Saline + CIAP);(6)Rats pretreated with warfarin administered at a dose of 90 µg/kg/dose before induction of cerulein-induced AP (WF 90 + CIAP);(7)Rats pretreated with warfarin administered at a dose of 180 µg/kg/dose before induction of cerulein-induced AP (WF 180 + CIAP);(8)Rats pretreated with warfarin administered at a dose of 270 µg/kg/dose before induction of cerulein-induced AP (WF 270 + CIAP).

Warfarin (Warfin, Orion Corporation, Espoo, Finland) was administered at doses of 90, 180, or 270 µg/kg/dose (in groups 2–4 and 6–7) intragastrically once a day for 7 days consecutively before the induction of AP. Saline (in group 1 and 5) was administered in the same manner. Doses of warfarin were selected according to the doses used in people. Clinically, the initial dose of warfarin in adults is 5–10 mg/day. For a 50 kg patient, they are administered 100–200 µg/kg/day. For a 75 kg patient, they are administered 66.6–133.3 μg/kg/day. In the case of a 100 kg patient, they are administered 50–100 μg/kg/day.

Cerulein powder (Sigma-Aldrich, GmbH, Steinheim, Germany) was dissolved in saline to prepare the infusion solution, which was used to induced acute pancreatitis. Subsequently, cerulein was administered intraperitoneally 6 times within one-hour intervals, at a dose of 50 μg/kg per injection. Simultaneously, the animals from groups without the induction of AP (groups 1–4) were treated intraperitoneally with saline.

### 2.2. Measurement of the Pancreatic Blood Flow

Immediately after the last intraperitoneal injection of cerulein or saline, the rats were anesthetized with 50 mg/kg of ketamine (i.p., Bioketan, Vetoquinol Biowet, Gorzów Wielkopolski, Poland) and the experiment was finished. The animals were weighed, and the abdominal cavity was open to visualize the pancreas; the blood flow was checked using a laser Doppler flowmeter (PeriFlux 4001 Master Monitor, Perimed AB, Järfälla, Sweden). Blood flow was measured in different portions of the pancreas, and by dint of the probe’s area and depth of the measurement, the method determined the total pancreatic blood flow. This method has been described previously in detail [61,62]. Data were presented as a percent change from the value obtained in control saline-treated rats without the induction of AP.

### 2.3. Biochemical Analysis

After the assessment of the pancreatic blood flow, blood samples were taken from the abdominal aorta for biochemical analysis.

The prothrombin time, expressed as the international normalized ratio (INR), was checked in the fresh blood using Alere INRatio^®^ 2 PT/INR Monitoring Systems and Alere INRatio^®^ PT/INR Monitoring System Test Strips (Alere San Diego, Inc., San Diego, CA, USA).

The plasma D-dimer concentration was measured using an immunoturbidimetric assay (Innovance D-Dimer Assay, Siemens Healthcare GmbH, Marburg, Germany) on the automatic coagulation analyzer BCS XP System (Siemens Healthcare Diagnostics, Erlangen, Germany). The results are presented in µg/mL.

Serum lipase and amylase activity was determined with a Kodak Ectachem DT II System analyzer (Eastman Kodak Company, Rochester, NY, USA) using Lipa and Amyl DT Slides (Vitros DT Chemistry System, Johnson & Johnson Clinical Diagnostic, Inc., Rochester, NY, USA). The results are presented in U/L.

The serum level of interleukin-1β (IL-1β) was measured using the Rat IL-1β Platinum Elisa (Bender MedSystem GmbH, Vienna, Austria). This kit is an in vitro enzyme-linked immunosorbent assay for the quantitative measurement of rat interleukin—1β. The results are presented in pg/mL.

### 2.4. Measurement of the Pancreatic Weight and Pancreatic Histology

After blood collection, the pancreas was cut out from the body and weighed, and then its weight was calculated per 100 g body weight. Specimens of pancreatic tissue were collected for histological examination. The samples were fixed in 10% buffered formaldehyde, embedded in paraffin, and sections were sliced and stained with hematoxylin and eosin. Slides were evaluated by two experienced pathologists without knowledge of the treatment provided. As described previously [63], the following histological signs of pancreatic damage were examined (grading from 0 to 3):Pancreatic edema: 0 = no edema, 1 = interlobular edema, 2 = interlobular and moderate intralobular edema, and 3 = severe interlobular and intralobular edema.Hemorrhages: 0 = absent, 1 = from one to two foci per slide, 2 = from three to five foci per slide, and 3 = more than five foci per slide.Leukocyte infiltration: 0 = absent, 1 = scarce perivascular infiltration, 2 = moderate perivascular and scarce diffuse infiltration, and 3 = abundant diffuse infiltration.Acinar necrosis: 0 = absent, 1 = less than 15% of cells involved, 2 = from 15% to 35% of cells involved, and 3 = more than 35% of cells involved.Vacuolization of acinar cells: 0 = absent, 1 = less than 25%, 2 = 25–50%, and 3 = more than 50%.

The results of the histological examination were exhibited as a predominant histological score of each sign of pancreatic damage in each experimental group.

### 2.5. Statistical Analysis 

Statistical analysis was conducted using an analysis of variance, followed by Turkey’s multiple comparison test using GraphPadPrism (GraphPad Software, San Diego, CA, USA). The results were presented as means ± standard error of the mean (SEM). There were eight animals in each of the experimental groups. A *p* value lower than 0.05 was considered statistically significant. 

## 3. Results

### 3.1. International Normalized Ratio

Prothrombin time was measured as the international normalized ratio (INR). In control rats receiving intragastric and intraperitoneal saline without the induction of AP (C), the INR reached a value of 1.1 ± 0.1 (Figure 1). In rats without the induction of AP, intragastric warfarin administered for 7 consecutive days at doses of 90, 180, and 270 µg/kg/dose (WF 90 + Saline, WF 180 + Saline, and WF 270 + Saline) led to a significant increase in the INR to values of 2.82 ± 0.22, 4.11 ± 0.28, and 6.38 ± 0.35, respectively.

In rats pretreated with saline before the induction of AP (Saline + CIAP), the INR was 1.5 ± 0.18. In rats pretreated with warfarin and the induction of AP (WF + CIAP), the INR levels were similar to those observed in the group pretreated with warfarin without the induction of AP (WF + Saline) (Figure 1).

### 3.2. Serum Concentration of D-Dimer

In control animals treated with intragastric and intraperitoneal saline (C), the plasma D-Dimer concentration reached a value 0.3 ± 0.04 (Figure 2). The intragastric administration of warfarin at the doses used (WF 90 + Saline, WF 180 + Saline, and WF + Saline) was without a significant effect on the plasma concentration of D-Dimer in rats without the induction of AP. The induction of AP in rats without the administration of warfarin (Saline + CIAP) led to more than a 9-fold increase in the plasma D-Dimer concentration. Pretreatment with warfarin at all the doses used (WF 90 + CIAP, WF 180 + CIAP, and WF 270 + CIAP) resulted in a partial but statistically significant reduction in the plasma D-Dimer concentration in AP rats (Figure 2).

### 3.3. Pancreatic Weight

Pancreatic weight is expressed in g per 100 g of body weight. In control rats treated with intragastric and intraperitoneal saline without the induction of AP (C), the main weight of the pancreas was 313 ± 11 mg/100 g of body weight (Figure 3). The administration of warfarin at doses of 90, 180, or 270 µg/kg/dose without the induction of AP (WF 90 + Saline, WF 180 + Saline, and WF 270 + Saline) had no significant effect on the weight of the pancreas/100 g of body weight. The induction of AP in rats pretreated with saline (Saline + CIAP) increased the pancreatic weight/100 g of body weight by 70% in comparison to a value observed in control animals (C). Pretreatment with warfarin administered at doses of 90 or 180 µg/kg/dose significantly reduced the pancreatitis-induced increase in pancreatic weight/100 g of body weight (WF 90 + CIAP and WF 180 + CIAP). In contrast, warfarin administered at a dose of 270 µg/kg/dose before the induction of AP (WF 270 + CIAP) resulted in an increase in pancreatic weight/100 g of body weight above that observed in saline-treated AP animals. Moreover, pancreatic weight/100 g of body weight in the group WF 270 plus AP was significantly higher than in the group treated with warfarin administered at a dose of 90 µg/kg/dose before the induction of AP (Figure 3).

### 3.4. Histological Examination 

In control saline-treated rats without the induction of AP, the pancreas showed no tissue alteration under light microscopic examination (Table 1, Figure 4(1)). In rats pretreated with warfarin at the doses used, without the induction of pancreatitis, no inflammatory infiltration, vacuolization of acinar cells, and foci of necrosis were observed in the pancreas. Pancreatic edema in histological images was not observed in rats without pancreatitis and pretreated with warfarin at a dose of 180 µg/kg/day, while in the case of a dose of 90 or 270 µg/kg/dose, no pancreatic edema was observed in some animals, but slight interlobular edema was observed in others. In half of the animals pretreated with warfarin at a dose of 270 µg/kg/dose, single hemorrhagic foci were observed; in the remaining animals from this group, no hemorrhagic foci were found in the pancreas (Table 1, Figure 4(2–4)).

The intraperitoneal administration of cerulein in rats pretreated with saline led to the development of acute edematous pancreatitis in all rats. Histological images of the pancreases showed moderate or severe interlobular and intralobular edema accompanied with moderate perivascular and scarce diffuse leukocytic infiltration. Vacuolization was observed in 25 to more than 50% of acinar cells. Necrosis or foci of hemorrhages were not observed (Table 1, Figure 4(5)). Pretreatment with warfarin administered at a dose of 90 or 180 µg/kg/dose prior to AP induction inhibited the development of morphological signs of pancreatic damage observed in histological examination (Table 1, Figure 4(6,7)). Pancreatic edema was reduced to interlobular or interlobular and moderate intralobular edema. Inflammatory infiltration was reduced to scare perivascular or moderate perivascular and scare the diffused infiltration of the pancreas. Vacuolization was observed in 25–50% of acinar cells. In contrast, in the case of animals pretreated with warfarin administered at a dose of 270 µg/kg/dose before the induction of AP, warfarin did not reduce the pancreatitis-evoked pancreatic edema, and this also caused the appearance of hemorrhagic foci in the pancreas (Table 1, Figure 4(8)).

### 3.5. Serum Activity of Pancreatic Enzymes

The serum activity of amylase in control saline-treated rats without the induction of AP (C) was 996 ± 80 U/L (Figure 5). In rats pretreated with warfarin without the induction of AP (WF + Saline), the serum activity of amylase was similar to that observed in control animals. The induction of AP by the administration of cerulein (Saline + CIAP) led to an almost 10-fold increase in the serum activity of amylase. Pretreatment with warfarin administered at a dose of 90 or 180 µg/kg/dose (WF 90 + CIAP and WF 180 + CIAP) partly but significantly reversed the AP-induced increase in the serum activity of amylase. This effect was more pronounced after warfarin administered at a dose of 180 µg/kg/dose. In rats pretreated with warfarin administered at a dose of 270 µg/kg/dose prior to the induction of AP (WF 270 + CIAP), the activity of serum amylase was even higher than in animals with AP pretreated with saline, but this effect was not statistically significant (Figure 5).

In control saline-treated rats without the induction of AP (C), the serum activity of lipase was 52.2 ± 5.2 U/L (Figure 6). In rats without the induction of AP, pretreatment with warfarin at the doses administered was without a significant effect on the serum activity of lipase (WF 90 + Saline, WF 180 + Saline, and WF 270 + Saline). The induction of AP by the intraperitoneal administration of cerulein (Saline + CIAP) led to an almost 9-fold increase in the serum activity of lipase. Pretreatment with warfarin administered at doses of 90 or 180 µg/kg/dose (WF 90 + CIAP and WF 180 + CIAP) tended to reverse the AP-induced increase in the serum activity of lipase; however, this effect was statistically unsignificant. In contrast, pretreatment with warfarin administered at a dose of 270 µg/kg/dose (WF 270 + CIAP) did not even tend to reduce the serum activity of lipase in rats with AP (Figure 6).

### 3.6. Serum Concentration of Interleukin-1β

The serum level of pro-inflammatory interleukin-1β (IL-1β) in control saline-treated rats without the induction of AP (C) was 83.0 ± 5.1 pg/mL (Figure 7). In rats without the induction of AP, pretreatment with warfarin caused a slight increase in the serum level of IL-1β, and this effect was statistically significant in rats pretreated with warfarin administered at doses of 180 and 270 µg/kg/dose (WF 180 + Saline and WF 270 + Saline). In rats pretreated with warfarin administered at a dose of 90 µg/kg/dose (WF 90 + Saline), this effect was not observed. In rats pretreated with saline, the induction of AP by cerulein led to a 3-fold increase in the serum level of IL-1β (Saline + CIAP). Pretreatment with warfarin administered at doses of 90 or 180 µg/kg/dose significantly reduced the pancreatitis-evoked increase in the serum level of IL-1β (WF + CIAP and WF 180 + CIAP). Pretreatment with warfarin administered at a dose of 270 µg/kg/dose was without any marked effect on the serum level of IL-1β in rats with AP (WF 270 + CIAP) (Figure 7).

### 3.7. Pancreatic Blood Flow

In control rats without the induction of AP and pretreated with intragastric and intraperitoneal saline, pancreatic blood flow reached a value of 100 ± 6.9% (Figure 8). In rats without the induction of AP, the administration of warfarin at the doses used was without a significant effect on pancreatic blood flow. The induction of AP by intraperitoneal cerulein reduced pancreatic blood flow by about 48%. Pretreatment with warfarin administered at doses of 90 or 180 µg/kg/dose tended to reverse the pancreatitis-evoked reduction in pancreatic blood flow. In contrast, pretreatment with warfarin administered at a dose of 270 µg/kg/dose led to an additional but insignificant reduction in pancreatic blood flow in rats with AP. Moreover, pancreatic blood flow in rats treated with WF 270 before the induction of AP was significantly lower than in rats treated with WF 90 before the induction of AP (Figure 8).

## 4. Discussion

As indicated in the introduction, there is a bidirectional relationship between inflammation and coagulation [1,2,3,4,5,6,7], and both those processes contribute to the severity and morality of acute pancreatitis [30,31,33,34]. Previous studies have shown that the inhibition and/or modification of coagulation by the administration of unfractionated heparin [38,39,40,44], low-molecular-weight heparin (LMWH), or recombinant human soluble thrombomodulin [48] inhibits the development and reduced the severity of AP.

In this study we examined the effect of pretreatment with warfarin, one of the coumarins, on the development of cerulein-induced AP in rats. Cerulein administration results in mild edematous acute pancreatitis, and the mechanism of this inflammation is primary-vascular-independent [64]. We provided functional, biochemical, and histological evidence that pretreatment with low doses of warfarin can prevent the development of AP evoked by cerulein.

Pancreatic ischemia and disturbances in pancreatic microcirculation play an important role in the pathogenesis of pancreatic injury in AP [65,66,67]. Reduced or insufficient blood flow triggers many mechanisms that play a key role in the development and course of this disease, such as the activation of leukocytes and coagulation, and the intrapancreatic release of pancreatic digestive enzymes and pro-inflammatory cytokines [65,66,68]. Primary local capillary leakage promotes the development of systemic inflammatory response syndrome (SIRS), multiple organ disfunction syndrome (MODS) [69], acute lung injury [70], disseminated intravascular coagulation (DIC) [33,34], and, in severe cases, may result in the patient’s death [71]. 

On the other hand, the improvement in blood supply prevents the development of damage to organs and accelerates their recovery.

On the other hand, the improvement in blood flow prevents the development of damage to organs and accelerates their recovery. In the digestive system, this effect has been observed in the oral cavity [61], stomach [72,73,74], duodenum [73], colon [75,76,77,78,79], and pancreas [39,40,47,60,75,80,81]. In our current study, cerulein-induced AP was found to worsen pancreatic blood flow, and treatment with warfarin administered at doses of 90 or 180 µg/kg/dose significantly improved it. This finding indicates that one of the protective mechanisms of warfarin in AP is to improve blood supply to the pancreas, but the exact mechanism is not known. This is probably related to the anticoagulant activity of this coumarin derivative.

The history of coumarin dates back to 1820, when Vogel for the first time extracted it from tonka bean [82]. Warfarin, one of the coumarins derivatives, is a racemic mixture of R- and S-enantiomers and acts by inhibiting epoxide reductase, thereby interfering with the hepatic synthesis of vitamin-K-dependent mature clotting factors II, VII, IX, and X and proteins C and S [83,84]. By inhibiting the conversion of inactive oxidized vitamin K epoxide to its active, reduced hydroquinone form [85], the drug ceases coagulation cascade in the intrinsic and extrinsic pathway. In addition, the use of vitamin K antagonists increases the permeability of the clot with the enhanced efficiency of fibrinolysis [86]. Warfarin owns stereospecific pharmacodynamic properties, and the isomers are heterogeneously metabolized by cytochrome p450 [84,87] and present a dose-dependent effect. Due to its longer half-life (36 h), it is used more frequently than acenocoumarol (10 h) [58]. Moreover, warfarin has been shown to maintain an INR within more stable ranges [88], and also has better anticoagulant properties [89] compared to acenocoumarol. On the basis of our experiments, we found that pretreatment with warfarin administered at doses of 90 or 180 µg/kg/dose inhibited the development of cerulein-induced AP. Our study confirmed that AP development significantly increases the INR and plasma D-Dimer concentration. This indicates that in the course of mild, edematous AP, the coagulation is activated, and thrombi are formed in the circulation, followed by fibrinolysis [28,32]. In our experiment, warfarin increased the INR in a dose-dependent manner, and the INR ranges were similar in rats with and without the subsequent induction of AP. Moreover, in animals pretreated with warfarin, the concentration of D-Dimer in plasma was significantly lower after AP induction than in the control group. This observation permits an assumption that pretreatment with warfarin reduces the level of clotting factors and diminishes the activation of coagulation during AP induction. This led to a reduction in clot formation, and thus reduced the level of fibrin degradation products.

In our present study, serum amylase and lipase activity after the induction of AP were, respectively, almost 10-fold and 9-fold higher than in the control group. The administration of warfarin at doses of 90 or 180 µg/kg/dose for 7 consecutive days prior to AP induction partly but significantly reversed the AP-induced increase in the serum activity of amylase and lipase. This seems to be both a result and a mechanism of the protective effect of coumarin derivative on the pancreas. It is widely recognized that elevated levels of amylase and lipase have a high sensitivity and specificity in the diagnosis of AP [90,91]. It was reported that the presence of active pancreatic digestive enzymes in circulation leads to an increase in the expression of adhesive molecules on endothelial cells and leukocytes, which results in impaired pancreatic blood flow through leukocyte–endothelial interaction [92]. As we have already discussed, abnormal pancreatic microcirculation plays an important role in the mechanism of AP. These data indicate that lowering the activity of pancreatic enzymes in the blood of animals receiving warfarin led to a decrease in the direct damaging effect of these enzymes on tissues, as well as reduced inflammatory infiltration, which together resulted in the inhibition of AP development.

A histological examination of the pancreatic tissue showed that pretreatment with warfarin administered at doses of 90 or 180 µg/kg/dose significantly reduced the morphological signs of AP evoked by cerulein. The degree of pancreatic edema, inflammatory infiltration, and vacuolization of acinar cells declined in those groups. A reduction in pancreatic edema in groups treated with warfarin at doses of 90 or 180 µg/kg/dose before the induction of AP was also found in the macroscopic examination, as well as a reduction in the pancreatic weight of AP animals. In contrast, pretreatment with warfarin at a dose of 270 µg/kg/dose before the induction of AP did not decrease the AP-induced pancreatic edema in the microscopic and macroscopic examinations, and neither did it reduce pancreatic weight gain in AP animals. In addition, this high dose of warfarin combined with the induction of AP led to the development of hemorrhagic foci in the pancreas in some animals. This lack of the protective effects of warfarin administered at a dose of 270 µg/kg/day on pancreatic morphology, as well as the lack of the inhibition of the increase in pancreatic weight or serum activity of the amylase and serum concentration of IL-1β, appears to be due to the excessive inhibition of coagulation. In rats treated with warfarin administered at a dose of 270 µg/kg/day, the INR was above six. The observed changes in pancreatic histology clearly indicate that during the use of warfarin, it is necessary to monitor the INR value, and the INR value should be maintained within the range of 2–3.5. This range of the INR is recommended in most clinical indications for anticoagulant treatment [93].

Acinar cells, stellate cells, and resident macrophages secrete chemokines and cytokines such as interleukin-8 (IL-8), monocyte chemotactic protein-1 (MCP-1, known also as CCL2 (C-C motif chemokine ligand 2), CCL 5 (C-C motif chemokine ligand 5) also known as RANTES (regulated on activation, normal T cell expression, and secretion)) and CCL 3 or MLP-1, and they are involved in the activation of leukocyte adhesion molecules, integrins, and the transendothelial migration of leukocytes to the damaged area [94,95]. Moreover, proinflammatory cytokines such as interleukin-1β (Il-1β), tumor necrosis factor-α (TNFα), platelet-activating factor (PAF), and interleukin-6 (Il-6) are major contributors to the development of inflammation and systemic inflammatory response (SIRS) and multiple organ dysfunction (MODS), leading to the development of severe acute pancreatitis [95,96,97,98]. IL-6 acts to promote the production of acute-phase proteins such as fibrinogen, C-reactive protein, amyloid A, and hepcidin, while reducing the synthesis of albumin and cytochrome P450 in hepatocytes [99]. It was shown that the level of Il-6 and Il-1β corresponds with the severity of acute pancreatitis [100] and the overexpression of Il-1β in the murine induces chronic pancreatitis [101]. In the present study, we showed that pretreatment with warfarin before evoking cerulein-induced pancreatitis results in a decrease in the serum concentration of interleukin-1β and a limitation of the severity of AP. These important observations confirm the previous reports that acenocoumarol administration in an ischemic and edematous model of AP and the administration of warfarin in ischemia-reperfusion-induced pancreatitis significantly decrease the serum IL-1 level and led us to conclude that coumarin derivatives provide universal protection before AP [55,56,60].

Our present study showed, on the basis of biochemical, histological, and functional evidence, that pretreatment with warfarin at doses of 90 and 180 µg/kg/dose before the induction of AP by cerulein administration inhibits the development of this disease. From a clinical point of view, this preventive therapy matters for patients undergoing procedures burdened with the risk of developing acute pancreatitis, such as endoscopic retrograde cholangiopancreatography (ERCP). Currently, approximately 350,000–500,000 of ERCPs are performed annually in the United States alone [102,103], illustrating how many patients can benefit from the successful prevention of AP development.

Hypertriglyceridemia, which is one of the most common lipid abnormalities encountered in clinical practice, is associated, among others, with obesity, and is also one of the most common causes of AP [104]. It also constitutes as a risk factor for critical care admission in patients with AP [105]. One of the diagnostic options for hypertriglyceridemia-induced pancreatitis is heparin and insulin infusion, which significantly decreases the triglyceride level [42,43,106]; however, no clinical guidelines for anticoagulant therapy exist to date.

In AP, in addition to damage to the pancreas, other organs, such as the kidneys and lungs, are also damaged. In addition, the course of acute pancreatitis is age-dependent. In older patients and animals, the course of acute pancreatitis is much more severe than in young beings [107]. These relationships are also associated with significantly increased thrombosis in the lungs and kidneys and a marked increase in the concentration of plasminogen activator inhibitor-1 (PAI-1) in the course of AP in the late period of life [107]. These observations suggest the direction of future research, as well as the possibility of improving the course of AP in elderly patients as a result of anticoagulant therapy.

Moreover, there are reports that preexisting anticoagulation can protect from the development of AP [108]. Of course, anticoagulant therapy carries the risk of effects, among which the risk of bleeding is particularly important. Warfarin is one of the most common causes of emergency hospitalization among the elderly due to adverse drug events, which is due to the narrow therapeutic window of the drug [109]. A strong dietary impact on the INR is also well known [110]. However, it was proofed that standardized warfarin monitoring decreases adverse drug reactions [111]. Moreover, the effect of warfarin can be reversed by the administration of vitamin K, fresh frozen plasma, prothrombin complex concentrate, or, in exceptional clinical cases, recombinant activated factor VII (rFVIIa) [112,113]. In recent years, it was shown that the cytochrome P450 (CYP2C9) and vitamin K epoxide reductase complex 1 (VKORC1) genotype is a major determinant of warfarin dose requirements and also impacts the risk of over-anticoagulation and hemorrhage, especially at the beginning of therapy [114]. For this reason, genetic counselling and genetic testing to guide warfarin dosage seems to be reasonable [115] and improves the safety of the therapy.

## 5. Conclusions

Our results indicate that pretreatment with warfarin inhibits the development of cerulein-induced acute pancreatitis (AP). This observation confirms the concept that the activation of coagulation plays an important role in the development of AP independently of the primary cause of this disease. In addition, our observations suggest that action aimed at modifying the activity of the coagulation system in the course of AP may contribute significantly to the therapy of this disease.

## Figures and Tables

**Figure 1 biomolecules-13-00948-f001:**
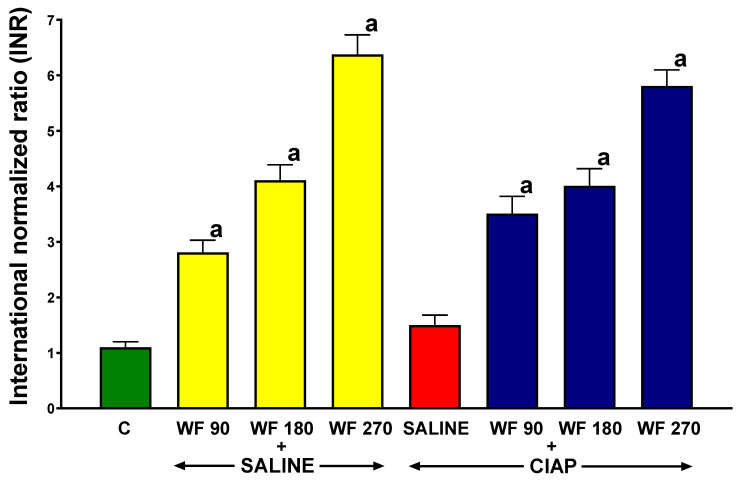
Effect of intragastric pretreatment with warfarin or saline on the prothrombin time measured as international normalized ratio (INR) in rats receiving intraperitoneal saline or rats with acute pancreatitis (cerulein-induced acute pancreatitis—CIAP). Key: C = saline-treated control; CIAP = cerulein-induced acute pancreatitis; WF = warfarin; 90 = 90 μg/kg/day; 180 = 180 μg/kg/day; 270 = 270 μg/kg/day. Mean ± SEM. N = 8 in each group of rats. ^a^
*p* < 0.05 compared to control.

**Figure 2 biomolecules-13-00948-f002:**
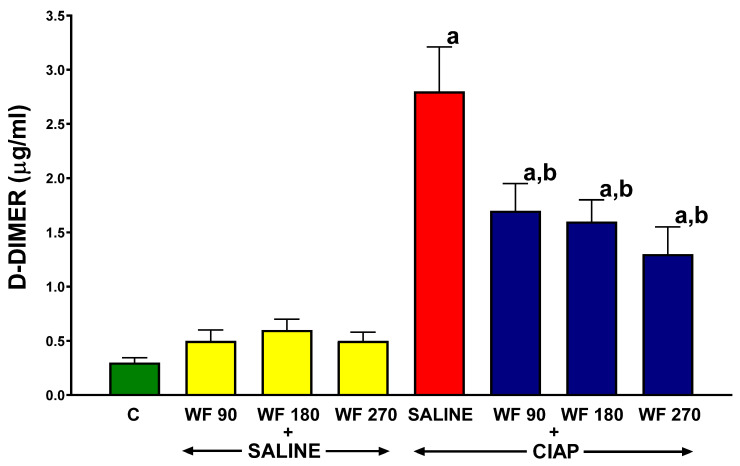
Effect of intragastric pretreatment with warfarin or saline on plasma D-Dimer concentration in rats receiving intraperitoneal saline or rats with acute pancreatitis (cerulein-induced acute pancreatitis—CIAP). Key: C = saline-treated control; CIAP = cerulein-induced acute pancreatitis; WF = warfarin; 90 = 90 μg/kg/day; 180 = 180 μg/kg/day; 270 = 270 μg/kg/day. Mean ± SEM. N = 8 in each group of rats. ^a^
*p* < 0.05 compared to control; ^b^
*p* < 0.05 compared to saline + CIAP.

**Figure 3 biomolecules-13-00948-f003:**
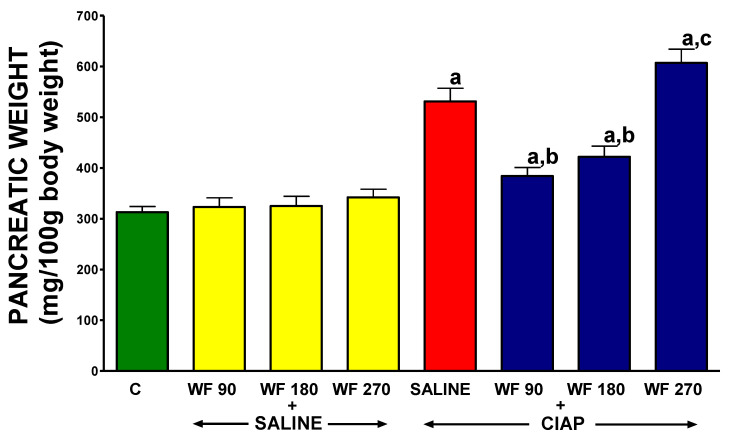
Effect of intragastric pretreatment with warfarin or saline on pancreatic weight in rats receiving intraperitoneal saline or rats with acute pancreatitis (cerulein-induced acute pancreatitis—CIAP). Key: C = saline-treated control; CIAP = cerulein-induced acute pancreatitis; WF = warfarin; 90 = 90 μg/kg/day; 180 = 180 μg/kg/day; 270 = 270 μg/kg/day. Mean ± SEM. N = 8 in each group of rats. ^a^
*p* < 0.05 compared to control; ^b^
*p* < 0.05 compared to saline + CIAP; ^c^
*p* < 0.05 compared to WF 90 + CIAP.

**Figure 4 biomolecules-13-00948-f004:**
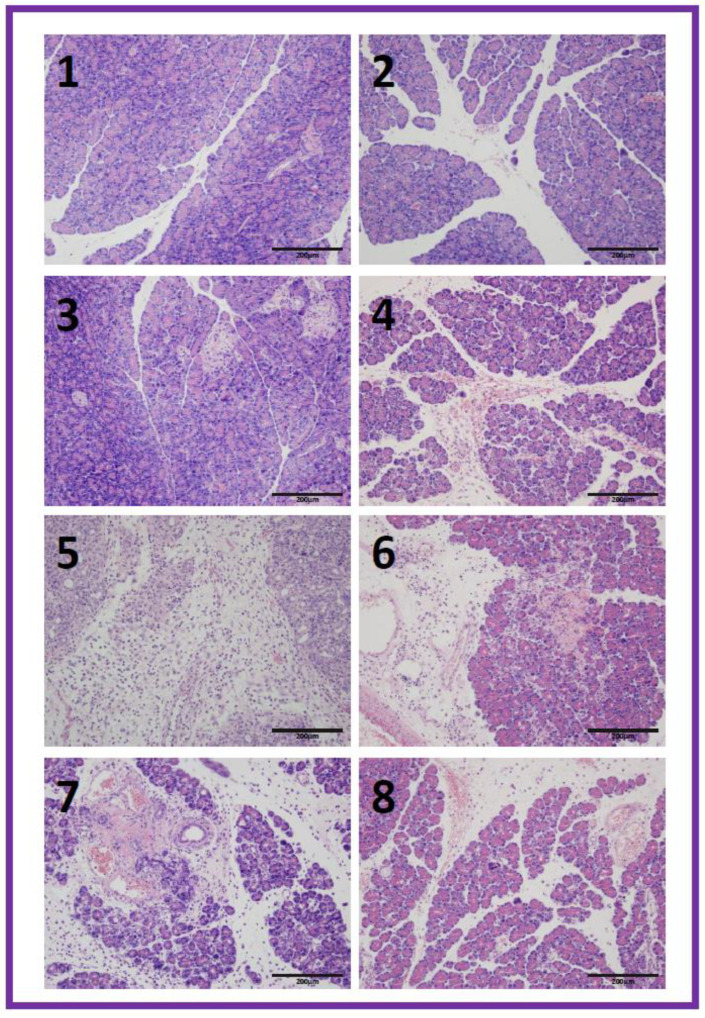
Representative morphological features of the pancreas observed in saline-treated control rats (**1**); rats pretreated with warfarin at a dose of 90 µg/kg/day without induction of acute pancreatitis (**2**); rats treated with warfarin at a dose of 180 µg/kg/day without induction of acute pancreatitis (**3**); rats treated with warfarin at a dose of 270 µg/kg/day without induction of acute pancreatitis (**4**); rats with cerulein-induced acute pancreatitis pretreated with saline (**5**); rats pretreated with warfarin at a dose of 90 µg/kg/day plus cerulein-induced acute pancreatitis (**6**); rats pretreated with warfarin at a dose of 180 µg/kg/day plus cerulein-induced acute pancreatitis (**7**); rats pretreated with warfarin at a dose of 270 µg/kg/day plus cerulein-induced acute pancreatitis (**8**). Hematoxylin–eosin stain, original magnification 200×.

**Figure 5 biomolecules-13-00948-f005:**
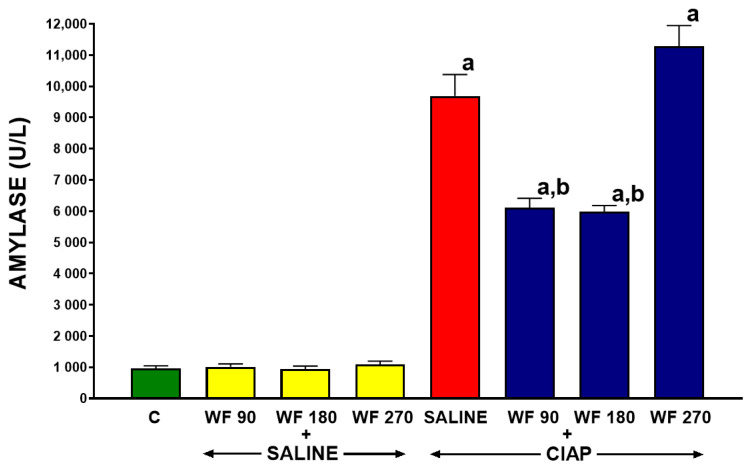
Effect of intragastric pretreatment with warfarin or saline on serum activity of amylase in rats receiving intraperitoneal saline or rats with acute pancreatitis (cerulein-induced acute pancreatitis—CIAP). Key: C = saline-treated control; CIAP = cerulein-induced acute pancreatitis; WF = warfarin; 90 = 90 μg/kg/day; 180 = 180 μg/kg/day; 270 = 270 μg/kg/day. Mean ± SEM. N = 8 in each group of rats. ^a^
*p* < 0.05 compared to control; ^b^
*p* < 0.05 compared to saline + CIAP.

**Figure 6 biomolecules-13-00948-f006:**
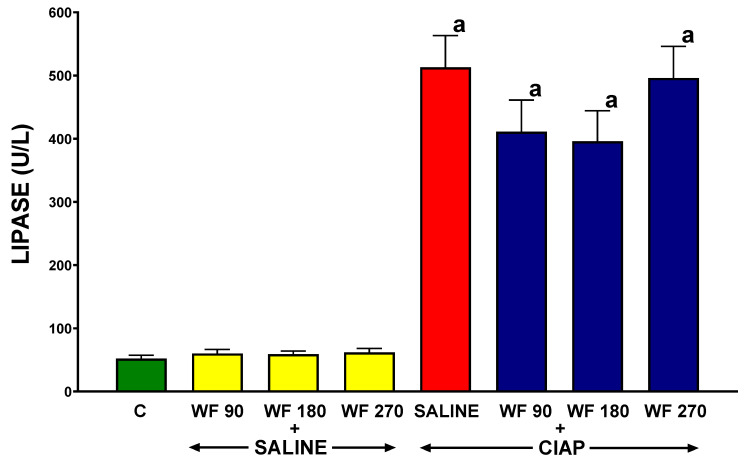
Effect of intragastric pretreatment with warfarin or saline on serum activity of lipase in rats receiving intraperitoneal saline or rats with acute pancreatitis (cerulein-induced acute pancreatitis—CIAP). Key: C = saline-treated control; CIAP = cerulein-induced acute pancreatitis; WF = warfarin; 90 = 90 μg/kg/day; 180 = 180 μg/kg/day; 270 = 270 μg/kg/day. Mean ± SEM. N = 8 in each group of rats. ^a^
*p* < 0.05 compared to control.

**Figure 7 biomolecules-13-00948-f007:**
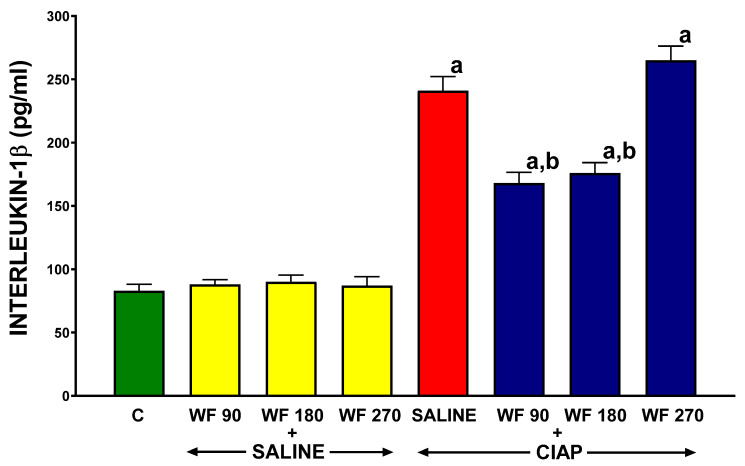
Effect of intragastric pretreatment with warfarin or saline on serum concentration of interleukin-1β in rats receiving intraperitoneal saline or rats with acute pancreatitis (cerulein-induced acute pancreatitis—CIAP). Key: C = saline-treated control; CIAP = cerulein-induced acute pancreatitis; WF = warfarin; 90 = 90 μg/kg/day; 180 = 180 μg/kg/day; 270 = 270 μg/kg/day. Mean ± SEM. N = 8 in each group of rats. ^a^
*p* < 0.05 compared to control; ^b^
*p* < 0.05 compared to saline + CIAP.

**Figure 8 biomolecules-13-00948-f008:**
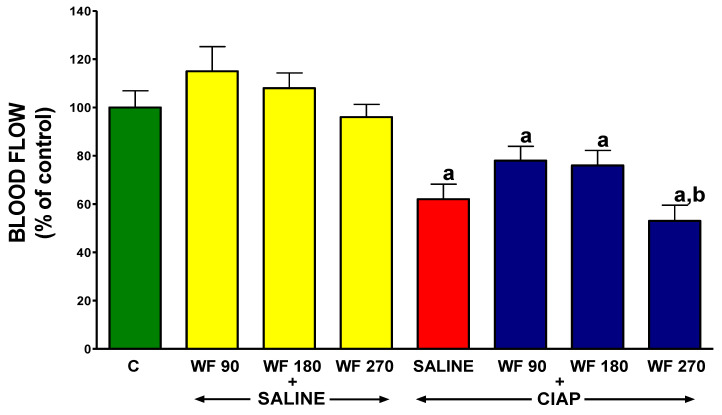
Effect of intragastric pretreatment with warfarin or saline on pancreatic blood flow in rats receiving intraperitoneal saline or rats with acute pancreatitis (cerulein-induced acute pancreatitis—CIAP). Key: C = saline-treated control; CIAP = cerulein-induced acute pancreatitis; WF = warfarin; 90 = 90 μg/kg/day; 180 = 180 μg/kg/day; 270 = 270 μg/kg/day. Mean ± SEM. N = 8 in each group of rats. ^a^
*p* < 0.05 compared to control; ^b^
*p* < 0.05 compared to WF 90 + CIAP.

**Table 1 biomolecules-13-00948-t001:** Effect of cerulein-induced acute pancreatitis (CIAP) and pretreatment with warfarin administered at doses of 90, 180, or 270 μg/kg/day, applied alone or in their combination (warfarin plus CIAP) on morphological signs of pancreatic damage.

	EDEMA (0–3)	INFLAMMATORY INFILTRATION (0–3)	VACUOLIZATION (0–3)	NECROSIS (0–3)	HEMORRHAGES (0–3)
CONTROL	0	0	0	0	0
WARFARIN 90 + saline	0–1	0	0	0	0
WARFARIN 180 + saline	0	0	0	0	0
WARFARIN 270 + saline	0–1	0	0	0	0–1
CIAP	2–3	2	2–3	0	0
WARFARIN 90 + CIAP	1–2	1–2	2	0	0
WARFARIN 180 + CIAP	1–2	1	2	0	0
WARFARIN 270 + CIAP	2–3	1–2	2	0	1

Numbers represent the predominant histological grading in each group.

## Data Availability

Data available on request due to restrictions e.g., privacy or ethical. The data presented in this study are available on request from the corresponding author. The data are not publicly available due to privacy restrictions.

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
