# Peer review of "Administration of Warfarin Inhibits the Development of Cerulein-Induced Edematous Acute Pancreatitis in Rats"

_biomolecules, 2023, doi:10.3390/biom13060948_

Round 1
Reviewer 1 Report
Authors have been focused in this article on the effects of warfarin in cerulein–induced acute pancreatitis (AP). AP is a sudden inflammation of the pancreas. The incidence of pancreatitis is increasing and AP has recently become the most common reason of hospitalisation with high mortality rate amongst gastrointestinal diseases in the United States. In spite of the numerous efforts during the last decades to explore the etiology, pathomechanism and course of pancreatitis, a specific disease-modifying therapy is still missing, therefore the topic is relevant. However, numerous experimental and clinical data confirm the effectiveness of heparin in the treatment of AP.
Criticism:
1. Description of the methods is very poor. It must to describe this part of the manuscript with more details.
2. Pancreas weight at the time of euthanasia is an easy-to-measure and useful parameter to judge edematous or atrophic pancreas responses, but it is depend on the body weight of the animal. It would be more correct to use pancreas weight/body weight ratio to evaluate the severity of the disease.
3. It would be useful to present also pictures from the tissue with lower magnification to evaluate the histological changes and to visualize the present results in a bar chart with individual data. Scale bar is missing from all pictures, it should be replace.
4. Authors show that 90 and 180 µg/kg/day warfarin pretreatment has a protective role in the pathogenesis of AP, however 270 µg/kg/day has no effect. What do the authors think, what is the reason of this observation?
5. Additionally, 90 and 270 µg/kg/day of warfarin caused decided difference in interlobular edema compared to untreated animals. What is the explanations for this?
6. Also IL-6 (and other cytokine as well) is corresponds with the severity of AP, but the authors measured just the level of IL-1β. It should more informative to measure another inflammatory cytokines as well.
7. What do the authors think whether warfarin with other administration protocol (posttreatment or administration during induction of AP) would be effective in the development and treatment of AP?
Minor comments:
1. The significance (between CIAP salin vs. CIAP WF 90 and WF 180) is missing on Figure 6 and Figure 8. It should be replace.
2. 59 and 80 reference is the same……
3. The following text on page 14 is repeated word for word in the summary……..
„Acinar cells, stellate cells and resident macrophages secrete chemo[1]kines and cytokines such as interleukin-8 (IL-8), monocyte chemotactic protein-1 (MCP-1, known also as CCL 2, C-C motif chemokine ligand 5 (CCL 5) also known as RANTES (regulated on activation, normal T cell expressed and secreted) and CCL 3 or MLP-1 involved in the activation of leukocyte adhesion molecules, integrins and transendothelial migration of leukocytes to the damaged area [93,94]”
Author Response
Response to comments of Reviewer 1
Authors have been focused in this article on the effects of warfarin in cerulein–induced acute pancreatitis (AP). AP is a sudden inflammation of the pancreas. The incidence of pancreatitis is increasing and AP has recently become the most common reason of hospitalisation with high mortality rate amongst gastrointestinal diseases in the United States. In spite of the numerous efforts during the last decades to explore the etiology, pathomechanism and course of pancreatitis, a specific disease-modifying therapy is still missing, therefore the topic is relevant. However, numerous experimental and clinical data confirm the effectiveness of heparin in the treatment of AP.
Response: We thank the reviewer for valuable comments and suggestions regarding our manuscript, as well as the opinion that the topic of our research is relevant. We have substantially revised our manuscript according to the reviewer’s suggestions. Referring to the reviewer’s comments, we prepared point-by-point responses to each comment of the reviewer, and we have marked changes in the text of the manuscript in yellow.
Criticism:
1.Description of the methods is very poor. It must to describe this part of the manuscript with more details.
Response: Additional details on the methodology are provided in the new version of the manuscript.
2.Pancreas weight at the time of euthanasia is an easy-to-measure and useful parameter to judge edematous or atrophic pancreas responses, but it is depend on the body weight of the animal. It would be more correct to use pancreas weight/body weight ratio to evaluate the severity of the disease.
Response: According to the reviewer comment, in the new version of the manuscript, the weight of the pancreas is presented per 100 g of body weight. Changes in the text of results are highlighted in yellow.
3.It would be useful to present also pictures from the tissue with lower magnification to evaluate the histological changes and to visualize the present results in a bar chart with individual data. Scale bar is missing from all pictures, it should be replace.
Response: In the new version of the manuscript the scale in graphic form has been added in all images.
4.Authors show that 90 and 180 µg/kg/day warfarin pretreatment has a protective role in the pathogenesis of AP, however 270 µg/kg/day has no effect. What do the authors think, what is the reason of this observation?
Response: In the current version of the manuscript we have added the following statement in the Discussion: “This lack of protective effects of warfarin given at the dose of 270 µg/kg/day on pancreatic morphology, as well as the lack of inhibition of the increase in pancreatic weight or serum activity of amylase and serum concentration of IL-1β appear to be due to excessive inhibition of coagulation. In rats treated with warfarin given at a dose of 270 µg/kg/day, the INR was above 6. The observed changes in pancreatic histology clearly indicate that during the use of warfarin it is necessary to monitor the INR value, and the INR value should maintained within range of 2-3.5. This range of INR is recommended in most clinical indications for anticoagulant treatment (Baczynska A. Doustne leki przeciwkrzepliwe w róznych stanach klinicznych – praktyczny poradnik. Choroby Serca i NaczyÅ„ 2004; 1: 27-36).”
5.Additionally, 90 and 270 µg/kg/day of warfarin caused decided difference in interlobular edema compared to untreated animals. What is the explanations for this?
Response: See response to comment 4.
6.Also IL-6 (and other cytokine as well) is corresponds with the severity of AP, but the authors measured just the level of IL-1β. It should more informative to measure another inflammatory cytokines as well.
Response: In our current study, we used a model cerulein-induced pancreatitis in rats. In this model, mild acute pancreatitis develops and the degree of pancreatitis severity is assessed shortly after the onset of pancreatitis. Early studies have shown that IL-1β is the first a key interleukin released in the inflammatory cascade. IL-1β acts directly as well as by stimulating the release of other members of the pro-inflammatory cascade. IL-1b plays an essential role in the induction of systemic acute phase response by stimulating the production of proinflammatory cytokines such as interleukin-6 (IL-6), stimulating the synthesis of adhesion molecules in endothelial cells and leukocytes, promoting thrombocytosis, pyrogen release, and production of acute phase proteins such as C-reactive protein (PMID: 8630372; PMID: 19302047; PMID: 9286248). Other pro-inflammatory cytokines, including IL-6, appear later after pancreatitis is more fully developed. Therefore, in our research, we focused on the assessment of IL-1β levels.
7.What do the authors think whether warfarin with other administration protocol (posttreatment or administration during induction of AP) would be effective in the development and treatment of AP?
Response: It is a good idea. We have already conducted a study on the therapeutic effect of warfarin in the course of cerulein-induced pancreatitis. The obtained results confirms this effect and the manuscript.
Minor comments:
1.The significance (between CIAP salin vs. CIAP WF 90 and WF 180) is missing on Figure 6 and Figure 8. It should be replace.
Response: We are very grateful for detecting this error in our manuscript. Unfortunately, the error did not occur on Figures, but in the description of the results. The error was due to the fact that statistical analysis and description of the results were done by different co-authors. The description of results has been corrected to match Figures. In addition, after statistical re-analysis, it was decided to mark the statistically significant difference between WF 90 plus CIAP and WF 270 plus CIAP in Figure 8, which was not marked in the previous version of the manuscript. Changes made to the text are highlighted in yellow.
2.59 and 80 reference is the same……
Response: Thank you for your comment. Reference 80 has been replaced by other article (PMID: 17622699).
3.The following text on page 14 is repeated word for word in the summary……..
„Acinar cells, stellate cells and resident macrophages secrete chemo[1]kines and cytokines such as interleukin-8 (IL-8), monocyte chemotactic protein-1 (MCP-1, known also as CCL 2, C-C motif chemokine ligand 5 (CCL 5) also known as RANTES (regulated on activation, normal T cell expressed and secreted) and CCL 3 or MLP-1 involved in the activation of leukocyte adhesion molecules, integrins and transendothelial migration of leukocytes to the damaged area [93,94]”
Response: Thanks for pointing out the repetitive sentences. They have been removed from the current version of the manuscript.
Reviewer 2 Report
This is a manuscript entitled “Administration of warfarin inhibits the development of caerulein-induced edematous acute pancreatitis in rats” by Katarzyna Konarska-Bajda et al., a qualified research team in the field. The study is to test the authors’ hypothesis that anticoagulant treatment may be an effective therapy for acute pancreatitis. The same research groups had previously shown that pretreatment of rats with low molecular weight heparin can successfully prevent development of acute pancreatitis in rats. Thus, the current study with use of warfarin, as an extension of this former study, is expected to show a similar result. To induce experimental acute pancreatitis, caerulein at a dose of 50 micrograms per kg body weight was intraperitoneally injected to rats with hourly interval. This is a widely used acceptable model of acute pancreatitis. The experimental design was to compare the efficacy of warfarin with three different doses (90, 180, or 270 micrograms per kg, daily for 7 days), which is reasonable. Blood flow rate, biochemical assay (for coagulation, inflammation, and pancreatic damages), and histological assessment were performed to compare the severity of the disease. Inclusion of such multiple analyses is appropriate. The researchers found that pretreatment with warfarin at doses of 90 and 80, but not 270, showed significant beneficial effects to prevent development of acute pancreatitis. Overall, this manuscript is very well written. Abstract and Introduction are well constructed and informative. Obtained data are well described in graphs which are all comprehensive. This reviewer has only a few comments which hopefully improve this manuscript.
(1) The rationale of the tested warfarin doses (90, 180, 270 ug/kg) should be inserted in Methodology section. Are these doses comparable to clinical doses for human?
(2) Acute pancreatitis causes organ damage and coagulation not only in pancreas but also in other organs such as lungs and kidneys (Okamura et al. Aging Cell 2012, PMCID: PMC3784256). Were tissue damage or coagulation examined in other organs?
(3) It is intriguing that the highest dose of warfarin (270 ug/kg) showed no beneficial effect. Optimization of warfarin dose is known to be tricky as the optimal dose is often different among individuals. Authors are encouraged to provide more discussion why the highest dose of warfarin had no effects.
Author Response
Response to comments of Reviewer 2
This is a manuscript entitled “Administration of warfarin inhibits the development of caerulein-induced edematous acute pancreatitis in rats” by Katarzyna Konarska-Bajda et al., a qualified research team in the field. The study is to test the authors’ hypothesis that anticoagulant treatment may be an effective therapy for acute pancreatitis. The same research groups had previously shown that pretreatment of rats with low molecular weight heparin can successfully prevent development of acute pancreatitis in rats. Thus, the current study with use of warfarin, as an extension of this former study, is expected to show a similar result. To induce experimental acute pancreatitis, caerulein at a dose of 50 micrograms per kg body weight was intraperitoneally injected to rats with hourly interval. This is a widely used acceptable model of acute pancreatitis. The experimental design was to compare the efficacy of warfarin with three different doses (90, 180, or 270 micrograms per kg, daily for 7 days), which is reasonable. Blood flow rate, biochemical assay (for coagulation, inflammation, and pancreatic damages), and histological assessment were performed to compare the severity of the disease. Inclusion of such multiple analyses is appropriate. The researchers found that pretreatment with warfarin at doses of 90 and 80, but not 270, showed significant beneficial effects to prevent development of acute pancreatitis. Overall, this manuscript is very well written. Abstract and Introduction are well constructed and informative. Obtained data are well described in graphs which are all comprehensive.
Response: We thank the reviewer 2 for valuable comments and suggestions regarding our manuscript, as well as the favorable opinion that the manuscript is well-written, abstract and Introduction are well constructed and informative, and obtained data are well described in graphs which are all comprehensive. We have revised our manuscript according to the reviewer’s suggestions. Referring to the reviewer’s comments, we prepared point-by-point responses to each comment of the reviewer, and we have marked changes in the text of the manuscript in yellow.
This reviewer has only a few comments which hopefully improve this manuscript.
(1) The rationale of the tested warfarin doses (90, 180, 270 ug/kg) should be inserted in Methodology section. Are these doses comparable to clinical doses for human?
Response: Doses of warfarin were selected according to the doses used in people. Clinically, initial dose of warfarin in adults is 5–10 mg/day. For the 50 kg patient, it gives 100–200 µg/kg/day. For the 75 kg patient, it gives 66.6–133.3 μg/kg/day. In the case of 100 kg patient, it gives 50–100 μg/kg/day. This statement is present in the current version of the manuscript in subsection 2.1. Animals and treatment.
(2) Acute pancreatitis causes organ damage and coagulation not only in pancreas but also in other organs such as lungs and kidneys (Okamura et al. Aging Cell 2012, PMCID: PMC3784256). Were tissue damage or coagulation examined in other organs?
Response: Unfortunately, our present study did not include the assessment of damage to other organs in the course of acute pancreatitis. But such studies could be important in evaluating the systemic effects warfarin administration. In the present manuscript, we can only mention such effects using the article suggested by the reviewer (see the penultimate paragraph in the Discussion).
(3) It is intriguing that the highest dose of warfarin (270 ug/kg) showed no beneficial effect. Optimization of warfarin dose is known to be tricky as the optimal dose is often different among individuals. Authors are encouraged to provide more discussion why the highest dose of warfarin had no effects.
Response: In the current version of the manuscript we have added the following statement in the Discussion: “This lack of protective effects of warfarin given at the dose of 270 µg/kg/day on pancreatic morphology, as well as the lack of inhibition of the increase in pancreatic weight or serum activity of amylase and serum concentration of IL-1β appear to be due to excessive inhibition of coagulation. In rats treated with warfarin given at a dose of 270 µg/kg/day, the INR was above 6. The observed changes in pancreatic histology clearly indicate that during the use of warfarin it is necessary to monitor the INR value, and the INR value should maintained within range of 2-3.5. This range of INR is recommended in most clinical indications for anticoagulant treatment (Baczynska A. Doustne leki przeciwkrzepliwe w róznych stanach klinicznych – praktyczny poradnik. Choroby Serca i NaczyÅ„ 2004; 1: 27-36).”
Round 2
Reviewer 1 Report
Thank you for the answer and the modifications. I have no addiitional comments.